# MC_MoveAbsolute() 4th Order Real-Time Trajectory Generation Function Algorithm and Implementation

**Krzysztof Pietrusewicz \***, **Paweł Waszczuk** and **Michał Kubicki**

Department of Industrial Automation and Robotics, West Pomeranian University of Technology, 70-310 Szczecin, Poland; pawel.waszczuk@zut.edu.pl (P.W.); michal.kubicki@zut.edu.pl (M.K.)

**\*** Correspondence: krzysztof.pietrusewicz@zut.edu.pl; Tel.: +48-663-398-396

**Featured Application: The theoretical results and practical implementation of the trajectory generator presented in this article are the outcome of two research projects: The first was issued by the Polish Ministry of Science and Higher Education and the second one was part of the European Union's 7th Framework Programme (see Funding Section).**

**Abstract:** This paper presents the issue of generating motion trajectories in a digital servo drive in accordance with the PLCopen Motion Control standard. This standard does not limit the details of motion generation in the electromechanical systems, but indicates its interface and set of necessary parameters. Moreover, it is placed within a state machine, which allows the individual software elements to integrate with it seamlessly. This work discusses time-optimal point-to-point trajecto-ries, i.e., the initial and final reference speeds are zero, and they are compliant with the MC_MoveAbsolute() function defined in the PLCopen Motion Control standard. The smoothness of the resulting trajectory can be attributed to the use of a fourth order trajectory generator, which defines the bounds up to snap – the second derivative of acceleration. One of the aims of this article was to bridge the theoretical aspect of trajectory generation with the algorithms practical implementation, by the means of PLC code generation using the MATLAB/Simulink package.

**Keywords:** PLCopen Motion Control; trajectory generation; digital servo drive; motion control; real-time algorithm implementation

---

## 1. Introduction

Servo drive systems are exploited widely in industries due to their efficiency, high-performance in motion control, and fast response times to dynamic reference signals [1,2]. In order to comply with the technological requirements, modern automatic machines have to guarantee decent dynamics for position and velocity control of motors, these servo drives solutions are utilized in Computer Numerically Controlled (CNC) machining, robotics, factory automation, and many other areas.

The most common control scheme implemented in servo drives to control position and velocity is the cascade control algorithm [3]. Its task is to regulate current, velocity, and position of the motor in separate closed control loops, in order to minimize overshoot and reduce the influence of external disturbances.

Nowadays, an increasing number of commercial control system manufacturers provide open architecture systems [4], meeting the need to constantly improve the servo drives functionality. This approach allows for design, testing, and integration of new control algorithms and techniques such as control laws, multi-sensor integration, or Artificial Intelligence (AI).

Simultaneously, evolution of the servo drive system programming languages is progressing. Currently, most of the commercial systems allow the user to implement custom code in the native

control structure. These modifications can be written in popular programming languages such as C, C++, or Structured Text (ST).

Automatic code generation [5] is one of the ideas which facilitates implementing the code in various industrial control platforms. It is based on text languages (C, ST) and allows to compile and transfer developed code or algorithms between devices of different classes and manufacturers with minimal effort on the programmer's part.

In the following work, generation of motion profiles (especially position) for servo drives is considered. This is a crucial aspect of applications that demand a high motion fidelity, e.g., CNC feed drives and multi-axis manipulators. In this case, constraining the acceleration profile obtained by the second order trajectory generator leads to oscillations whenever the movement direction changes (i.e., zero crossing of set velocity). This is the main reason for using third and higher order trajectory generators, for defining constraints on jerk or snap [6].

This paper is organized as follows: Section 1 serves as a brief introduction to control of digital servo drives; Section 2 is devoted to PLCopen Motion Control, particularly to one of its motion functions—MC_MoveAbsolute(). Section 3 presents a step by step algorithm to find a motion profile with boundaries defined up to the fourth derivative of position, and the trajectory generator of the fourth order is described. This topic is expanded upon in Section 4, where a real-time calculation of the beforementioned profile is discussed. Finally, Section 5 puts these solutions in context of the Programmable Logic/Automation Controller (PLC/PAC) devices that are IEC 61131-3 compliant, while Section 6 summarizes the paper.

## 2. Trajectory Generation in Digital Servo Drives: PLCopen Motion Control

In References [6–9], trajectory generation algorithms used in motion solutions are presented. The dynamics of mechanical aspects of the systems are considered in order to develop a feedforward controller. The following work is further expanding the ideas presented in Reference [9], such that the mathematical foundation laid there could be implemented in a real-time system.

In Reference [10], the concept of utilizing the PLCopen Motion Control standard for controlling the digital servo drive of the CNC machine is given. In the PLCopen Motion Control standard, several states are defined that outline the operation of a digital servo drive. These states depict the actual status of the motion system, being the type of movement executed (if any), present errors, stopping, or awaiting commands from the control system. Of particular interest are the two states tied to the motion execution:

(a)   Discrete Motion – movement to a defined position;
(b)   Continuous Motion – movement without a setpoint (e.g., shaft rotating with set velocity).

The transition between particular states is achieved by calling related functions, e.g., MC_MoveAbsolute (absolute movement) or MC_MoveAdditive (additive moment in a specified direction). The entire state machine is presented in Figure 1.

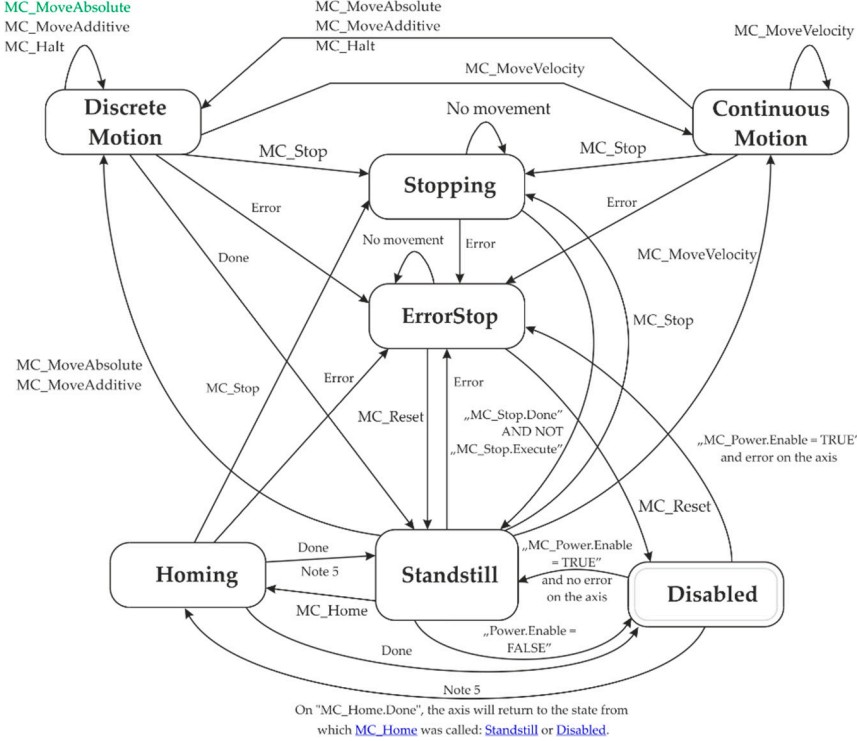

**Figure 1.** PLCopen Motion Control standard state execution state diagram.

*MC_MoveAbsolute() – Function Description*

This function block starts a controlled movement to a specified absolute position. All of the parameters needed to start the movement are transferred on a rising edge of the Execute input (Figure 2). The axis changes to the Discrete Motion state after all of them have been successfully transferred. Before the function block can be used for a real axis, the axis must be homed. States in which this function block can be used: Standstill, Discrete Motion, and Continuous Motion.

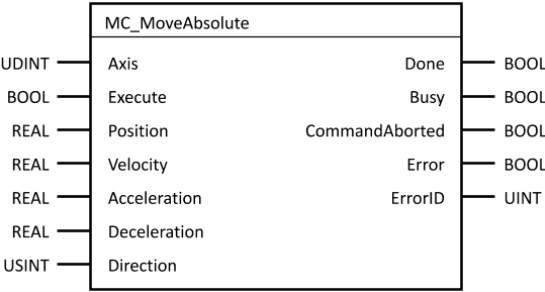

**Figure 2.** PLCopen Motion Control MC_MoveAbsolute()function block.

In the following figures, the timing diagrams for MC_MoveAbsolute() possible usage scenarios are shown. In Figure 3a, the first function block (MC_MoveAbsolute_0) stopped its movement before the second function block (MC_MoveAbsolute_1) was started. In Figure 3b, the second function block (MC_MoveAbsolute_1) was started before the first function block (MC_MoveAbsolute_0) completed its movement. In both cases, as seen in Figure 3a,b, the movements finish in the same position.

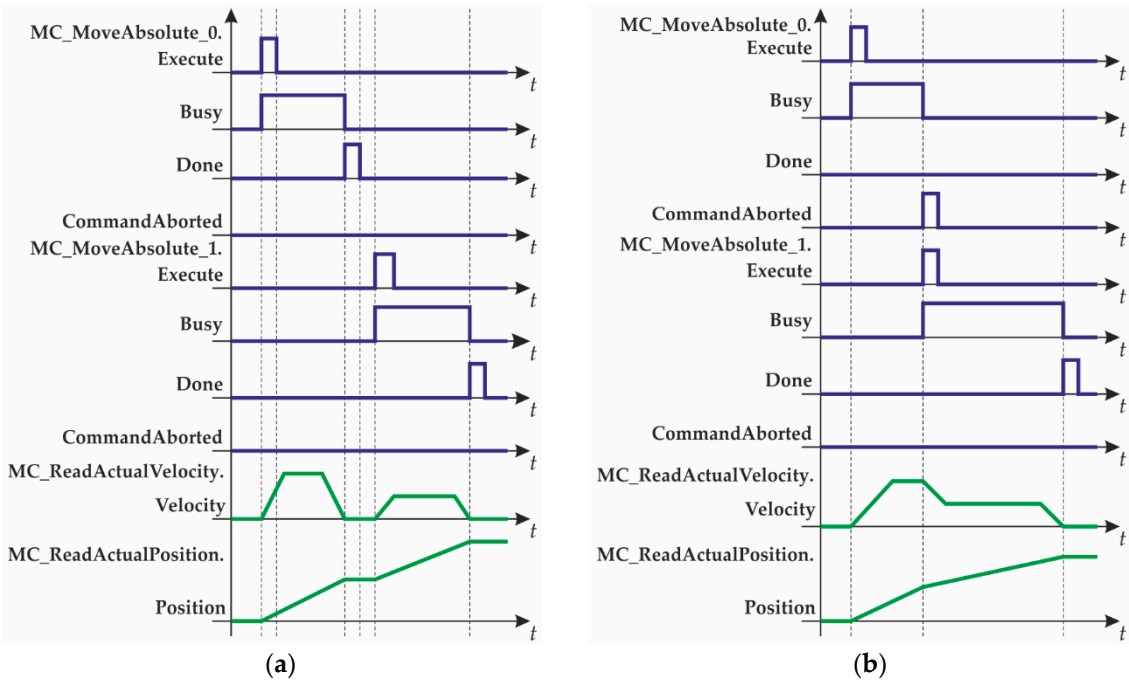

**Figure 3.** MC_MoveAbsoluteuse() example. (**a**): first use scenario; (**b**): second use scenario.

## 3. Fourth Order Position Profile Generation

In this work, the position profile will be understood as a setpoint trajectory with a sampling time specified for a real-time algorithm implementation. The following convention, adopted from [9], was used to denote the parameters: $x$ – position, displacement (profile, $m$); $v$ – velocity (profile, $m/s$); $a$ – acceleration (profile, $m/s^2$); $j$ – jerk (profile, $m/s^3$); $d$ – derivative of jerk (profile, $m/s^4$); $\overline{x}, \overline{v}, \overline{a}, \overline{j}, \overline{d}$ – bound on $|x|, |v|, |a|, |j|, |d|$, respectively; $t_{\overline{d}}, t_{\overline{j}}, t_{\overline{a}}, t_{\overline{v}}$ – time interval during in which $|d|, |j|, |a|, |v|$ obtains its bound, respectively; $T_s$ – sampling time in seconds.

The fourth order position trajectory generation problem was considered within this document. One input was used to define the desired position (i.e., increment from an actual position to the new one) and another four input parameters described the movement profile including velocity, acceler-ation, jerk, and snap. These latter parameters described the boundaries which cannot be crossed while executing the movement. The generated trajectory was time-optimal, i.e., the position was reached in the shortest possible time. For simplicity we assumed that the shape of the profiles was symmetric no matter what the direction of movement was.

$$\mathbf{u} = \left| \begin{bmatrix} \overline{x} & \overline{v} & \overline{a} & \overline{j} & \overline{d} \end{bmatrix}^T \right| \tag{1}$$

The direction of movement can be calculated based on the following equation:

$$sd = \mathrm{sgn}(\overline{x}) \tag{2}$$

where the $sd$ variable equals 1 if the movement direction is positive, or $-1$ when it is negative.

While considering the real-time implementation with respect to sampling time of code execu-tion, it is necessary to round up calculated time values:

$$t_{corr} = T_s \mathrm{ceil}\left(\frac{t_{calc}}{T_s}\right) \tag{3}$$

Fourth order trajectory profiles are shown in Figure 4.

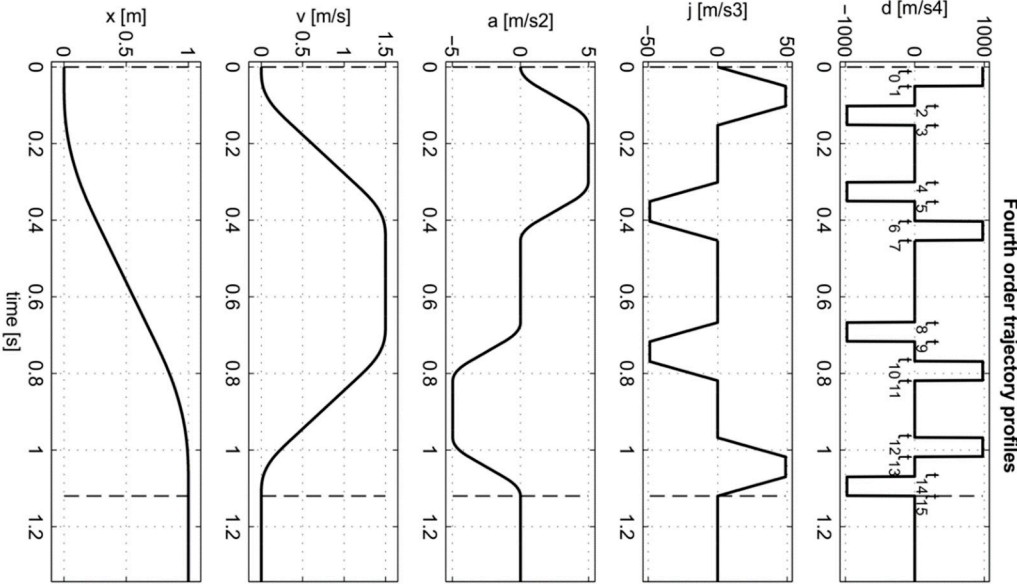

**Figure 4.** Fourth order trajectory profiles in point-to-point movement (MC_MoveAbsolute).

In the following sections, calculations for the type of trajectory generation for MC_MoveAbsolute() are shown in an algorithmic way.

### 3.1. Calculation of Time $t_d$

The following equations are used for calculation of time $t_{\overline{d}}$ which covers the duration of a single movement section with applied snap boundary (i.e., its constrained value), as in Figure 4.

$$t_{\overline{d}d_{\max}} = \sqrt[4]{\frac{\overline{x}}{8\overline{d}}} \tag{4}$$

Rounding up (4) with respect to the sampling time $T_s$:

$$T_s \mathrm{ceil}\left(\frac{t_{\overline{d}d_{\max}}}{T_s}\right) = T_s \mathrm{ceil}\left(\frac{\sqrt[4]{\frac{\overline{x}}{8\overline{d}}}}{T_s}\right) \tag{5}$$

$$dd = \left\lceil \frac{T_s \mathrm{ceil}\left(\frac{\sqrt[4]{\frac{\overline{x}}{8\overline{d}}}}{T_s}\right)}{8\left(T_s \mathrm{ceil}\left(\frac{\sqrt[4]{\frac{\overline{x}}{8\overline{d}}}}{T_s}\right)\right)^4} \right\rceil \tag{6}$$

If the following Equation (7) holds true:

$$\overline{v} \geq v_{\max} = 2\frac{\overline{x}}{8\left(T_s \mathrm{ceil}\left(\frac{\sqrt[4]{\frac{\overline{x}}{8\overline{d}}}}{T_s}\right)\right)^4}\left[T_s \mathrm{ceil}\left(\frac{\sqrt[4]{\frac{\overline{x}}{8\overline{d}}}}{T_s}\right)\right]^3 \tag{7}$$

Then:

$$dd1 = dd \tag{8}$$

Otherwise:

$$t_{\overline{d}v_{\max}} = \sqrt[3]{\frac{\overline{v}}{2\overline{d}}} \tag{9}$$

This rounded up with respect to the sampling time ($T_s$) yields the following equation:

$$T_s\text{ceil}\left(\frac{t_{\overline{d}v_{\max}}}{T_s}\right) = T_s\text{ceil}\left(\frac{\sqrt[3]{\frac{\overline{v}}{2\overline{d}}}}{T_s}\right) \tag{10}$$

Then:

$$dd1 = \left\lceil \frac{T_s\text{ceil}\left(\frac{\sqrt[3]{\frac{\overline{v}}{2\overline{d}}}}{T_s}\right)}{\frac{\overline{v}}{2\left(T_s\text{ceil}\left(\frac{\sqrt[3]{\frac{\overline{v}}{2\overline{d}}}}{T_s}\right)\right)^3}} \right\rceil \tag{11}$$

If the following Equation (12) holds true:

$$\overline{a} \geq a_{\max} = \frac{\overline{v}}{2\left(T_s\text{ceil}\left(\frac{\sqrt[3]{\frac{\overline{v}}{2\overline{d}}}}{T_s}\right)\right)^3}\left[T_s\text{ceil}\left(\frac{\sqrt[3]{\frac{\overline{v}}{2\overline{d}}}}{T_s}\right)\right]^2 \tag{12}$$

Then:

$$dd2 = dd1 \tag{13}$$

Otherwise:

$$t_{\overline{d}a_{\max}} = \sqrt[2]{\frac{\overline{a}}{\overline{d}}} \tag{14}$$

This rounded up with respect to the sampling time ($T_s$) yields the following equation:

$$T_s\text{ceil}\left(\frac{t_{\overline{d}a_{\max}}}{T_s}\right) = T_s\text{ceil}\left(\frac{\sqrt[2]{\frac{\overline{a}}{\overline{d}}}}{T_s}\right) \tag{15}$$

Then:

$$dd2 = \left\lceil \frac{T_s\text{ceil}\left(\frac{\sqrt[2]{\frac{\overline{a}}{\overline{d}}}}{T_s}\right)}{\frac{\overline{a}}{\left[T_s\text{ceil}\left(\frac{\sqrt[2]{\frac{\overline{a}}{\overline{d}}}}{T_s}\right)\right]^2}} \right\rceil \tag{16}$$

If the following Equation (17) holds true:

$$\overline{j} \geq j_{\max} = \frac{\overline{a}}{\left(T_s\text{ceil}\left(\frac{\sqrt[2]{\frac{\overline{a}}{\overline{d}}}}{T_s}\right)\right)^2}\left[T_s\text{ceil}\left(\frac{\sqrt[2]{\frac{\overline{a}}{\overline{d}}}}{T_s}\right)\right] \tag{17}$$

Then:

$$dd3 = dd2 \tag{18}$$

Otherwise:

$$t_{\overline{d}j_{\max}} = \frac{\overline{j}}{\overline{d}} \tag{19}$$

This rounded up with respect to the sampling time ($T_s$) yields the following equation:

$$T_s\text{ceil}\left(\frac{t_{\overline{d}j\text{max}}}{T_s}\right) = T_s\text{ceil}\left(\frac{\frac{\dot{j}}{\overline{d}}}{T_s}\right) \tag{20}$$

Then:

$$dd3 = \left[\begin{array}{c} \dfrac{T_s\text{ceil}\left(\frac{\frac{\dot{j}}{\overline{d}}}{T_s}\right)}{\overline{j}} \\[4mm] \dfrac{}{T_s\text{ceil}\left(\frac{\frac{\dot{j}}{\overline{d}}}{T_s}\right)} \end{array}\right] \tag{21}$$

Time $t_{\overline{d}}$ is derived from Equation (21) as:

$$t_{\overline{d}} = dd3_1 \tag{22}$$

The next section shows how to evaluate value of time ($t_{\overline{j}}$).

### 3.2. Calculation of Time $t_j$

When time ($t_{\overline{d}}$) is calculated, the very next step is to calculate the value of time ($t_{\overline{j}}$), while taking into account movement constraints and sampling time rounding of calculated values. The value $t_{\overline{j}}$ means the time of jerk (3rd order derivative of position), as shown in Figure 4. Derivation of time ($t_{\overline{j}}$) starts from calculations:

$$\begin{array}{rcl} P &=& -\frac{1}{9}t_{\overline{d}}^2 \\ Q &=& -\frac{1}{27}t_{\overline{d}}^3 - \frac{\overline{x}}{4dd_3[2]t_{\overline{d}}} \\ D &=& \sqrt{P^3 + Q^2} \\ D &=& \sqrt[3]{-Q + D} \\ t_{\overline{j}j\text{max}} &=& R - \frac{P}{R} - \frac{5}{3}t_{\overline{d}} \end{array} \tag{23}$$

This rounded up with respect to the sampling time ($T_s$) yields the following:

$$T_s\text{ceil}\left(\frac{t_{\overline{j}j\text{max}}}{T_s}\right) = T_s\text{ceil}\left(\frac{R - \frac{P}{R} - \frac{5}{3}t_{\overline{d}}}{T_s}\right) \tag{24}$$

Then:

$$dd4 = \left[\begin{array}{c} \dfrac{t_{\overline{d}}}{T_s\text{ceil}\left(\frac{t_{\overline{j}j\text{max}}}{T_s}\right)} \\[4mm] \dfrac{\frac{1}{2}}{t_{\overline{d}}\left\{4t_{\overline{d}}^3 + \left(T_s\text{ceil}\left(\frac{t_{\overline{j}j\text{max}}}{T_s}\right)\right)\left[8t_{\overline{d}}^2 + \left(T_s\text{ceil}\left(\frac{t_{\overline{j}j\text{max}}}{T_s}\right)\right)\left(5t_{\overline{d}} + T_s\text{ceil}\left(\frac{t_{\overline{j}j\text{max}}}{T_s}\right)\right)\right]\right\}} \end{array}\right] \tag{25}$$

If the following Equation (26) holds true:

$$\overline{v} \geq v'_{\text{max}} = dd4_1 dd4_3\left[2dd4_1{}^2 + dd4_2(3dd4_1 + dd4_2)\right] \tag{26}$$

Then:

$$dd5 = dd4 \tag{27}$$

Otherwise:

$$t_{\overline{j}v\text{max}} = \sqrt{\frac{1}{4}t_{\overline{d}}^2 + \frac{\overline{v}}{dd3_2 t_{\overline{d}}}} - 1.5t_{\overline{d}} \tag{28}$$

This rounded up with respect to the sampling time ($T_s$) yields the following:

$$T_s\text{ceil}\left(\frac{t_{\bar{j}v\text{max}}}{T_s}\right) = T_s\text{ceil}\left(\frac{\sqrt{\frac{1}{4}t_{\bar{d}}^2 + \frac{\bar{v}}{dd3_2 t_{\bar{d}}}} - 1.5t_{\bar{d}}}{T_s}\right) \tag{29}$$

Then:

$$dd5 = \left[\frac{t_{\bar{d}}}{\dfrac{T_s\text{ceil}\left(\frac{t_{\bar{j}v\text{max}}}{T_s}\right)}{t_{\bar{d}}\left\{2t_{\bar{d}}^2 + \left(T_s\text{ceil}\left(\frac{t_{\bar{j}v\text{max}}}{T_s}\right)\right)\left[3t_{\bar{d}} + T_s\text{ceil}\left(\frac{t_{\bar{j}v\text{max}}}{T_s}\right)\right]\right\}}}\right] \tag{30}$$

If the following Equation (31) holds true:

$$\bar{a} \geq a'_{\text{max}} = dd5_3 dd5_1 (dd5_1 + dd5_2) \tag{31}$$

Then:

$$dd6 = dd5 \tag{32}$$

Otherwise:

$$t_{\bar{j}a\text{max}} = \frac{\bar{a}}{dd3_2 t_{\bar{d}}} - t_{\bar{d}} \tag{33}$$

This rounded up with respect to the sampling time ($T_s$) yields the following:

$$T_s\text{ceil}\left(\frac{t_{\bar{j}a\text{max}}}{T_s}\right) = T_s\text{ceil}\left(\frac{\frac{\bar{a}}{dd3_2 t_{\bar{d}}} - t_{\bar{d}}}{T_s}\right) \tag{34}$$

Then:

$$dd6 = \left[\frac{t_{\bar{d}}}{\dfrac{T_s\text{ceil}\left(\frac{t_{\bar{j}a\text{max}}}{T_s}\right)}{t_{\bar{d}}\left[t_{\bar{d}} + T_s\text{ceil}\left(\frac{t_{\bar{j}a\text{max}}}{T_s}\right)\right]}}\right] \tag{35}$$

Time $t_{\bar{j}}$ is derived from Equation (35) as:

$$t_{\bar{j}} = dd6_2 \tag{36}$$

The following section considers calculations of acceleration phase time ($t_{\bar{a}}$).

### 3.3. Calculation of Time $t_a$

When times $t_{\bar{d}}$ and $t_{\bar{j}}$ are calculated, taking into account movement constraints and sam-pling time rounding of calculated values, the next step is to calculate value of time ($t_{\bar{a}}$). This means the time of acceleration (second order derivative of position), as shown in Figure 4. The derivation of time ($t_{\bar{a}}$) starts from the following calculations:

$$c = \left[\begin{array}{c} dd6_1(dd6_1 + dd6_2) \\ 3dd6_1\left\{2dd6_1{}^2 + dd6_2(3dd6_1 + dd6_2)\right\} \\ 2dd6_1\left\{dd6_1\left[4dd6_1{}^2 + dd6_2(8dd6_1 + 5dd6_2)\right] + dd6_2{}^3\right\} \end{array}\right] \tag{37}$$

And:

$$t_{\bar{a}a\text{max}} = \frac{1}{2}\frac{-c_2 + \sqrt{c_2{}^2 - 4c_1\left(c_3 - \frac{\bar{x}}{dd6_3}\right)}}{c_1} \tag{38}$$

This rounded up with respect to the sampling time ($T_s$) yields the following:

$$T_s \text{ceil}\left(\frac{t_{\bar{a}a_{\max}}}{T_s}\right) = T_s \text{ceil}\left(\frac{\frac{1}{2}\frac{-c_2+\sqrt{c_2{}^2-4c_1\left(c_3-\frac{\bar{x}}{dd6_3}\right)}}{c_1}}{T_s}\right) \tag{39}$$

Then:

$$dd7 = \begin{bmatrix} t_{\bar{d}} \\ t_{\bar{j}} \\ T_s\text{ceil}\left(\frac{t_{\bar{a}a_{\max}}}{T_s}\right) \\ \frac{\bar{x}}{\left(T_s\text{ceil}\left(\frac{t_{\bar{a}a_{\max}}}{T_s}\right)\right)\left[c_1\left(T_s\text{ceil}\left(\frac{t_{\bar{a}a_{\max}}}{T_s}\right)\right)+c_2\right]+c_3} \end{bmatrix} \tag{40}$$

If the following Equation (41) holds true:

$$\bar{v} \geq v''_{\max} = dd7_4 dd7_1\left[dd7_1(2dd7_1 + 3dd7_2 + dd7_3) + dd7_2(dd7_2 + dd7_3)\right] \tag{41}$$

Then:

$$dd8 = dd7 \tag{42}$$

Otherwise:

$$t_{\bar{a}v_{\max}} = \frac{dd6_1\left[dd6_1(-2dd6_1 - 3dd6_2) - dd6_2{}^2\right] + \frac{\bar{v}}{dd6_3}}{dd6_1(dd6_1 + dd6_2)} \tag{43}$$

This rounded up with respect to the sampling time ($T_s$) yields the following:

$$T_s\text{ceil}\left(\frac{t_{\bar{a}v_{\max}}}{T_s}\right) = T_s\text{ceil}\left(\frac{\frac{dd6_1\left[dd6_1(-2dd6_1-3dd6_2)-dd6_2{}^2\right]+\frac{\bar{v}}{dd6_3}}{dd6_1(dd6_1+dd6_2)}}{T_s}\right) \tag{44}$$

Then:

$$dd8 = \begin{bmatrix} t_{\bar{d}} \\ t_{\bar{j}} \\ T_s\text{ceil}\left(\frac{t_{\bar{a}v_{\max}}}{T_s}\right) \\ \frac{\bar{v}}{t_{\bar{d}}\left\{t_{\bar{d}}\left[2t_{\bar{d}}+3t_{\bar{j}}+T_s\text{ceil}\left(\frac{t_{\bar{a}v_{\max}}}{T_s}\right)\right]+t_{\bar{j}}\left[t_{\bar{j}}+T_s\text{ceil}\left(\frac{t_{\bar{a}v_{\max}}}{T_s}\right)\right]\right\}} \end{bmatrix} \tag{45}$$

Time $t_{\bar{a}}$ is derived from Equation (45) as:

$$t_{\bar{a}} = dd8_3 \tag{46}$$

The following section considers further calculations of the fourth order trajectory profile.

*3.4. Final Calculation of Movement Profile Times and Direction*

Final calculations (including time of constant velocity movement ($t_{\bar{v}}$)) start with:

$$t_{\bar{v}v_{\max}} = \frac{\bar{x} - dd8_4\left[t_{\bar{a}}(c_1 t_{\bar{a}} + c_2) + c_3\right]}{\bar{v}} \tag{47}$$

This rounded up with respect to the sampling time ($T_s$) yields the following:

$$T_s\text{ceil}\left(\frac{t_{\bar{v}v_{\max}}}{T_s}\right) = T_s\text{ceil}\left(\frac{\frac{\bar{x}-dd8_4\left[t_{\bar{a}}(c_1t_{\bar{a}}+c_2)+c_3\right]}{\bar{v}}}{T_s}\right) \tag{48}$$

Then:

$$
dd9 =
\begin{bmatrix}
t_{\overline{d}} \\
t_{\overline{j}} \\
t_{\overline{a}} \\
T_s \operatorname{ceil}\left( \frac{t_{\overline{v}} v_{\max}}{T_s} \right) \\
sd \dfrac{\overline{x}}{t_{\overline{a}}(c_1 t_{\overline{a}} + c_2) + c_3 + \left( T_s \operatorname{ceil}\left( \frac{t_{\overline{v}} v_{\max}}{T_s} \right) \right) t_{\overline{d}}\left[ t_{\overline{d}}\left( 2t_{\overline{d}} + 3t_{\overline{j}} + t_{\overline{a}} \right) + t_{\overline{j}}\left( t_{\overline{j}} + t_{\overline{a}} \right) \right]}
\end{bmatrix}
\tag{49}
$$

Where $sd$ is calculated in Equation (2).

The column vector in Equation (49) is the final part of the fourth order trajectory profile generation. Now we have all the time values necessary to create the movement profile based on the parameters given by the user. This part of the algorithm is called the trajectory planner. The next part of the algorithm is called the profile generator. It is described in the following section and its purpose is to generate a movement profile out of the times $t_{\overline{d}}, t_{\overline{j}}, t_{\overline{a}}, t_{\overline{v}}$ were calculated in previous sections.

## 4. Generation of Profile

The generation of a movement profile starts with the calculation of switching times $t_0 \ldots t_{15}$ as shown in Figure 4:

$$
t16 =
\begin{bmatrix}
0 & 0 & 0 & 0 \\
1 & 0 & 0 & 0 \\
1 & 1 & 0 & 0 \\
2 & 1 & 0 & 0 \\
2 & 1 & 1 & 0 \\
3 & 1 & 1 & 0 \\
3 & 2 & 1 & 0 \\
4 & 2 & 1 & 0 \\
4 & 2 & 1 & 1 \\
5 & 2 & 1 & 1 \\
5 & 3 & 1 & 1 \\
6 & 3 & 1 & 1 \\
6 & 3 & 2 & 1 \\
7 & 3 & 2 & 1 \\
7 & 4 & 2 & 1 \\
8 & 4 & 2 & 1
\end{bmatrix}
\begin{bmatrix}
t_{\overline{d}} \\
t_{\overline{j}} \\
t_{\overline{a}} \\
T_s \operatorname{ceil}\left( \frac{t_{\overline{v}} v_{\max}}{T_s} \right)
\end{bmatrix}
\tag{50}
$$

For simplicity, we can assume:

$$
t_{\overline{v}} \approx T_s \operatorname{ceil}\left( \frac{t_{\overline{v}} v_{\max}}{T_s} \right)
\tag{51}
$$

Then Equation (50) can be rewritten in the following form:

$$
t16 = \mathbf{S}_{M4}
\begin{bmatrix}
t_{\overline{d}} \\
t_{\overline{j}} \\
t_{\overline{a}} \\
t_{\overline{v}}
\end{bmatrix}
\tag{52}
$$

where $\mathbf{S}_{M4}$ describes the switching matrix for the fourth order trajectory profile generation. After these calculations, the following holds true as shown in Figure 5:

$$t16 = \begin{bmatrix} t_0 \\ t_1 \\ t_2 \\ t_3 \\ t_4 \\ t_5 \\ t_6 \\ t_7 \\ t_8 \\ t_9 \\ t_{10} \\ t_{11} \\ t_{12} \\ t_{13} \\ t_{14} \\ t_{15} \end{bmatrix} = \begin{bmatrix} 0 \\ t_{\bar{d}} \\ t_{\bar{d}} + t_{\bar{j}} \\ 2t_{\bar{d}} + t_{\bar{j}} \\ 2t_{\bar{d}} + t_{\bar{j}} + t_{\bar{a}} \\ 3t_{\bar{d}} + t_{\bar{j}} + t_{\bar{a}} \\ 3t_{\bar{d}} + 2t_{\bar{j}} + t_{\bar{a}} \\ 4t_{\bar{d}} + 2t_{\bar{j}} + t_{\bar{a}} \\ 4t_{\bar{d}} + 2t_{\bar{j}} + t_{\bar{a}} + t_{\bar{v}} \\ 5t_{\bar{d}} + 2t_{\bar{j}} + t_{\bar{a}} + t_{\bar{v}} \\ 5t_{\bar{d}} + 3t_{\bar{j}} + t_{\bar{a}} + t_{\bar{v}} \\ 6t_{\bar{d}} + 3t_{\bar{j}} + t_{\bar{a}} + t_{\bar{v}} \\ 6t_{\bar{d}} + 3t_{\bar{j}} + 2t_{\bar{a}} + t_{\bar{v}} \\ 7t_{\bar{d}} + 3t_{\bar{j}} + 2t_{\bar{a}} + t_{\bar{v}} \\ 7t_{\bar{d}} + 4t_{\bar{j}} + 2t_{\bar{a}} + t_{\bar{v}} \\ 8t_{\bar{d}} + 4t_{\bar{j}} + 2t_{\bar{a}} + t_{\bar{v}} \end{bmatrix} \tag{53}$$

Movement starts in time ($t_0$), while time moment ($t_{15}$) is the endpoint of the generated move-ment.

$$t_{final} = t16_{16} = 8t_{\bar{d}} + 4t_{\bar{j}} + 2t_{\bar{a}} + t_{\bar{v}} \tag{54}$$

Time values from Equation (53) are shown in the following Figure 5.

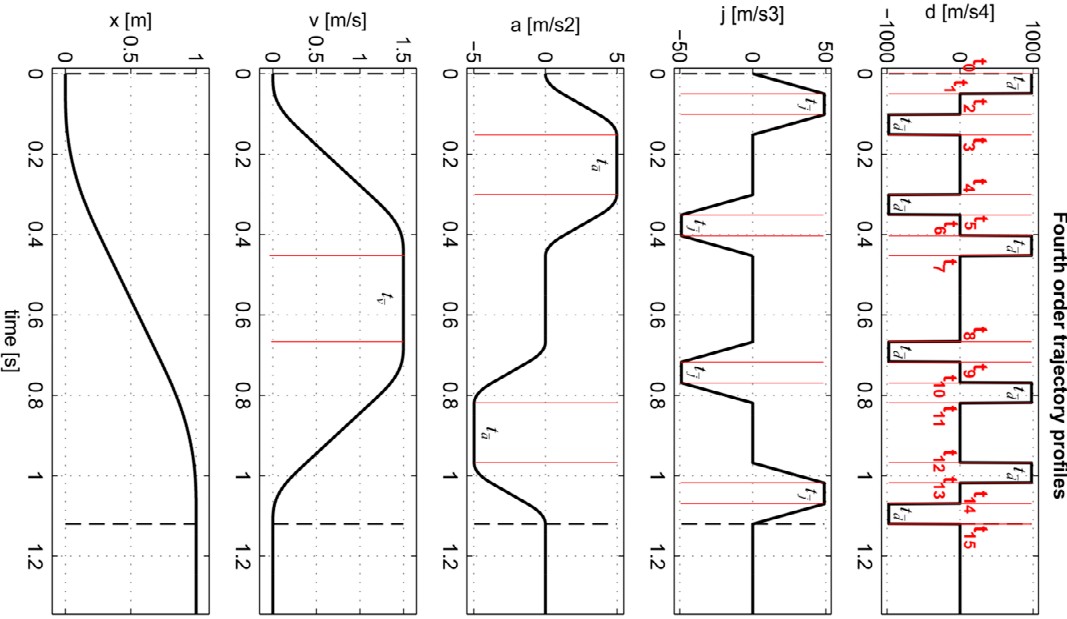

**Figure 5.** Fourth order trajectory profiles in *point-to-point* movement with profile times shown.

### 4.1. Execution of Trajectory Profile Generation

When the trajectory profile generation is executed (i.e., by the user) and output of the reference position trajectory has started, then the generation of snap profile is based on the following calcula-tions.

When:
$$start = \text{TRUE} \tag{55}$$

Then:
$$dd_{local} \leftarrow dd9_5 \tag{56}$$

The new value of $t_{16}$ is calculated based on Equation (52) and the resulting Equation (53). The variable *start* is set based on the algorithm shown in the following part of this subsection.

Until:
$$t_{exec} \leq t16_1 \tag{57}$$

Then:
$$p_1 = 0 \tag{58}$$

When:
$$t_{exec} > t16_1 \tag{59}$$

Then:
$$p_1 = dd9_5 \tag{60}$$

where $t_{exec}$ stands for the actual profile generation execution time. Calculations of $p_2 \ldots p_{16}$ values are based on the following conditions/equations:

$$
\begin{aligned}
&If\ t_{exec} \leq t16_2,\ p_2 = 0\\
&If\ t_{exec} > t16_2,\ p_2 = dd9_5
\end{aligned} \tag{61}
$$

Which gives the following algorithm:

$$
\begin{aligned}
&\text{FOR}\ j = 1 \ldots 16,\\
&If\ t_{exec} \leq t16_j,\ p_j = 0\\
&If\ t_{exec} > t16_j,\ p_j = dd9_5
\end{aligned} \tag{62}
$$

$p_1 \ldots p_{16}$ are calculated iteratively in each time sample of algorithm execution. Then the actual value/shape of derivative of jerk signal is calculated with the following equation:

$$
p_{d_{jerk}} = dir \left[
\begin{array}{l}
(p_1 - p_2 - p_3 + p_4) + \ldots \\
\ldots + (-p_5 + p_6 + p_7 - p_8) + \ldots \\
\ldots + (-p_9 + p_{10} + p_{11} - p_{12}) + \ldots \\
\ldots + (p_{13} - p_{14} - p_{15} + p_{16})
\end{array}
\right] \tag{63}
$$

where the *dir* variable stands for direction of movement. It is a two-valued variable:

$$dir = \{1, -1\} \tag{64}$$

calculated with Equations (65)–(68).

When the previous trajectory profile generation is finalized, i.e., time of execution ($t_{exec}$) is greater than the sum of $t_{final}\ [s]$ calculated with Equation (54) and some non-zero settling time ($T_{set}\ [s]$) has passed, then the system is ready for a new trajectory generation:

$$ready = \text{TRUE} \tag{65}$$

Determining direction of movement for Equation (63) starts with verifying the condition:

$$dir_{calc,k} = (dir > 0)\ AND\ (ready = \text{TRUE})\ AND\ NOT(enable = \text{TRUE}) \tag{66}$$

Input $EN = \text{TRUE}$ means that the user wishes for execution of the trajectory profile calculation (and profiled movement generation) and information about the readiness of the module for calculations is passed from the input variable *ready* to the variable *ENstart*. The rising edge of the condition:

$$(dir_{calc,k-1})OR(ENstart) \tag{67}$$

enables determination of direction, i.e., calculation of variable *dir*:

$$IF\,dir_{k-1} \leq 0\,THEN\,dir_k = -1\,ELSE\,dir_k = 1 \tag{68}$$

Equation (68) is used when we want to switch operation mode of the movement, i.e., motor/servo is mounted in an opposite direction.

### 4.2. Generation of Start Signal

Generation of the *start* signal that triggers Equation (56) and results in a new value of the $t_{16}$ variable (calculated based on Equation (52) and consequently Equation (53)) is connected with the equation:

$$IF\,(ENstart_k - ENstart_{k-1}) > 0\,THEN\,GenStart_k = 1 \tag{69}$$

The same *start* variable is used for triggering new profile calculations (Equations (1)–(53)).

$$start_k = GenStart_{k-1} \tag{70}$$

### 4.3. Calculation of Execution Time

Calculation of trajectory profiling execution time ($t_{exec}$) is based on the following expressions:

$$IF\,(GenStart_k \geq 0.5)\,THEN\,\left(t'_{exec,k} = -\frac{T_s}{2}\right)\,ELSE\,(t'_{exec,k} = t_{exec,k} + T_s)$$
$$t_{exec,k} = t'_{exec,k-1} \tag{71}$$

This finishes the algorithm of trajectory profile generation. The next step is to calculate actual values out of the process involving integrating snap signal values. It is shown in the next subsection.

### 4.4. Generation of Motion Profile Step

To output the motion profile of all the derivatives and the position itself, the following steps need to be executed in a real-time control platform:

- Reference signal $d_k$ (derivative of jerk/snap) is the zero-order-hold sampled value of $p_{d_{jerk}}$ variable.
- Reference signal $j_k$ (jerk) is the numerically integrated value of signal $d_k$ with respect to the sampling time ($T_s$):
$$j_k = T_s d_{k-1} + j_{k-1} \tag{72}$$

- Reference signal $a_k$ (acceleration) is the numerically integrated value of signal $j_k$ with respect to the sampling time ($T_s$):
$$a_k = T_s j_{k-1} + a_{k-1} \tag{73}$$

- Reference signal $v_k$ (velocity) is the numerically integrated value of signal $a_k$ with respect to the sampling time ($T_s$):
$$v_k = T_s a_{k-1} + v_{k-1} \tag{74}$$

- Reference signal $x_k$ (position) is the numerically integrated value of signal $v_k$ with respect to the sampling time ($T_s$):
$$x_k = T_s v_{k-1} + x_{k-1} \tag{75}$$

These steps finish the trajectory generation procedure.

## 5. Implementation of Trajectory Generator According to IEC 61131-3

The proposed algorithm was prepared to be transferrable to real-time implementation (e.g., PLC/PAC), which could later be used in industrial control platforms.

The algorithm was divided into the following sections:

- Plan Trajectory (Figure 6) – Equations (1)–(49);
- Generate profile (Figure 6) – Equations (50)–(71); while
- Generate Motion (Figure 6) subsystem calculates Equations (72)–(75).

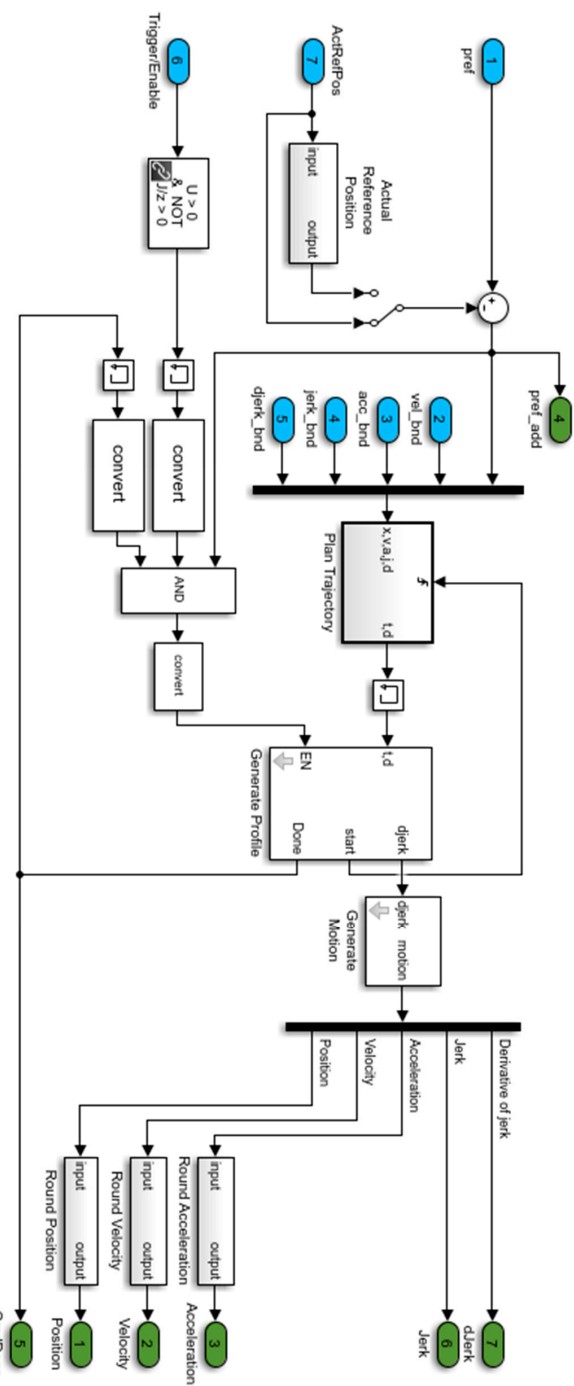

**Figure 6.** MATLAB/Simulink implementation of 4th order profile generator.

In Figure 7, two examples of generated trajectories are shown:

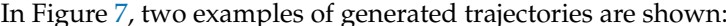

**Figure 7.** Two examples of generated trajectory profiles: (**a,b**) derivative of jerk, (**c,d**) jerk, (**e,f**) acceleration, (**g,h**) velocity, (**i,j**) position.

The function blocks code for the 4th order trajectory generator is presented in Figure 8, with the Simulink model presented in Figure 6 serving as a base for this conversion. Structured text code generated with a Simulink PLC Coder is about 900 lines long, and the total memory allocated for this function block in PLC is 3 kB. This means that the devised solution is implementable in the majority of PLC/PAC controllers which are IEC-61131-3 [11,12] standard compliant.

**Figure 8.** Code fragment generated via Simulink PLC Coder.

## 6. Summary

The article presents, in a step by step manner, the procedure for implementation of a 4th order trajectory generator that mirrors the MC_MoveAbsolute() function. Due to the straightforward nature of Equations (1)–(75), it is easily implementable in any of the available textual or graphical programming languages for real-time control systems.

The implementation of the presented algorithm is possible in control systems adhering to the IEC 61131-3 standard (e.g., PLC), and its source code was generated as a result of applying the fast prototyping paradigms, i.e., it was generated from the Simulink model. As an example, generated profiles are given with different parameters (i.e., boundaries). It is also possible to generate motion profiles of lower orders than fourth—the algorithm itself is scalable and following the procedure presented here will lead to obtaining simpler solutions constrained on third or second derivatives.

The most important contribution of this work is the unique nature of the presented approach, as it allows for practical implementation of this algorithm in commonly used real-time control systems.

The algorithm presented here is fully parametrized. It can be fully integrated by way of interfaces and its functionality, with other motion control features (i.e., collision avoidance or trajectory generation profile correction). As proposed in this paper, the implementation of *MC_MoveAbsolute*() is parametrized through its inputs (as it can be found in all functions and function blocks in graphical IEC 61131-3 compliant programming languages). Modularity of the approach enables the user to combine a proposed solution with an extension boom elasticity estimation feature (loader or forestry cranes applications) or dynamic collision avoidance (multi-axis serial robotic manipulators).

The function presented here can be implemented without any problems in all PLC/PAC-based control systems. The short sampling time (not more than 200 microseconds) requirement is fulfilled by most available control platforms. In the case of PLC/PACs with higher sampling times proposed, the algorithm can be extended with sampling time correction. In other cases, the integer multiple of sampling time requirement needs to be fulfilled by all times for $t_0$–$t_{15}$ times in Equation (53).

**Author Contributions:** Conceptualization, K.P. and M.K.; data curation, M.K.; formal analysis, P.W. and M.K.; funding acquisition, K.P.; investigation, K.P., P.W. and M.K.; methodology, K.P. and M.K.; project administration, K.P.; resources, K.P.; software, K.P.; supervision, K.P.; validation, P.W. and M.K.; visualization, P.W.; writing—original draft, K.P., P.W. and M.K.; writing—review and editing, K.P. and M.K.

**Funding:** This research was partially funded by: (a) Ministerstwo Nauki i Szkolnictwa Wyższego, grant number R03 042 02, (b) Ministerstwo Nauki i Szkolnictwa Wyższego, grant number N N502 336936 and, (c) Marie Curie 7PR, FP7-PEOPLE-2012-IAPP (Industry-Academia Partnerships and Pathways) grant number 324496.

**Conflicts of Interest:** The authors declare no conflict of interest.

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
