# Peer review of "MC_MoveAbsolute() 4th Order Real-Time Trajectory Generation Function Algorithm and Implementation"

_applsci, doi:10.3390/app9030538_

Reviewer 1 Report

The paper is of interest to readers who are involved in digital motion control applications. It presents a fourth order trajectory motion generator that has smooth derivatives upto the second derivative of acceleration. It also clearly shows how it can be implemented.

The major strength of this paper is not its novelty but that it shows how this technique can be implemented clearly using PLC motion control methods. It is also clearly written and presented.

Author Response

Dear Reviewer,

Thank you for your insightful analysis of our article. The article has been re-checked by an English lecturer. The minor linguistic corrections will certainly indicate proofreading of the final version (that is our experience with MDPI Publisher). By then we would like to leave the paper in its current form.

Best regards,

Krzysztof Pietrusewicz

Pietrusewicz Krzysztof, Associate Professor

Rector's representative for Industry 4.0

West Pomeranian University of Technology, Szczecin

Faculty of Electrical Engineering

Department of Control Engineering and Robotics

Sikorskiego 37, 70 – 313 Szczecin, Poland

Mobile: +48 663 398 396

Reviewer 2 Report

In general, it is an interesting publication, but it addresses concepts about trajectory planning that are widely known and published in the literature. It is recommended that authors introduce innovative and groundbreaking concepts that go beyond trajectory planning: for example introducing advanced control strategies, vibrational effects and their influence on trajectory planning, etc.

Author Response

Dear Reviewer,

Thank you for your insightful analysis of our article.

In general, the paper adresses concepts about trajectory planning that are widely known and published in the literature, however the novelty of the paper consists in a comprehensive presentation of the real-time implementation of a function compliant with PLCopen Motion Control MC_MoveAbsolute definition. That is something one cannot find in any published work, while usually it stands a trade secret of the motion control vendors. That is the main innovation of proposed here work.

The aim of this paper is to indicate the implementation of the fourth order trajectory generator. The authors are planning to present the issue of on-line optimization of trajectory generator parameters on the basis of collision avoidance algorithms in loading and unloading systems in forthcoming papers. Therefore, current paper stands a contribution to a series of publications in this area.

Best regards,

Krzysztof Pietrusewicz

Pietrusewicz Krzysztof, PhD, DSc, Associate Professor

Rector's representative for Industry 4.0

West Pomeranian University of Technology, Szczecin

Faculty of Electrical Engineering

Department of Control Engineering and Robotics

Sikorskiego 37, 70 – 313 Szczecin, Poland

Mobile: +48 663 398 396

Round  2

Reviewer 2 Report

Although, as mentioned by the authors, the trajectory planner is known, the details of implementation of the planner, that is, the algorithms for real time are secret, unravel and publish these algorithms has merits.

It is expected that the authors will continue with the following publications on collision detection and inherent aspects that have to do with security and contingencies that affect processes in real time.

It is an interesting and already acceptable article to be published. Considering that the conclusions are possibly one of the most important parts of a publication, the authors are asked to expand their conclusions on the aspects of loading and unloading collisions that will be implemented in the future and how this fits into the published algorithms. They could also expand their opinion about the implementation and technological scope of the presented developments in contrast to the modern servomotors and their hardware, for example.

Author Response

Dear Reviewer,

Again, thank you for your comments and remarks about our article.

The main topic of Michal's PhD thesis is trajectory generation with dynamic collision avoidance functionality. Presented here results will be a good reference starting point for his future scientific development.

Actual version of the paper includes answer to Your comment from the review.

Best regards,

Krzysztof Pietrusewicz

Pietrusewicz Krzysztof, Associate Professor

Rector's representative for Industry 4.0

West Pomeranian University of Technology, Szczecin

Faculty of Electrical Engineering

Department of Control Engineering and Robotics

Sikorskiego 37, 70 – 313 Szczecin, Poland

Mobile: +48 663 398 396